# Research on Lightweight Method of Insulator Target Detection Based on Improved SSD

**DOI:** 10.3390/s24185910

**Published:** 2024-09-12

**Authors:** Bing Zeng, Yu Zhou, Dilin He, Zhihao Zhou, Shitao Hao, Kexin Yi, Zhilong Li, Wenhua Zhang, Yunmin Xie

**Affiliations:** 1Nanchang Institute of Technology, Nanchang 330099, China; nathan997@163.com (D.H.); 13164162010@163.com (Z.Z.); 18331317390@163.com (S.H.); superstephen@sina.com (K.Y.); zhangwenhua_610@163.com (W.Z.); xie_yunmin@163.com (Y.X.); 2State Grid Shanghai Municipal Electric Power Company Maintenance Company, Shanghai 200063, China; ynotbn@sina.com

**Keywords:** SSD, insulator, lightweight, channel pruning, target detection

## Abstract

Aiming at the problems of a large volume, slow processing speed, and difficult deployment in the edge terminal, this paper proposes a lightweight insulator detection algorithm based on an improved SSD. Firstly, the original feature extraction network VGG-16 is replaced by a lightweight Ghost Module network to initially achieve the lightweight model. A Feature Pyramid structure and Feature Pyramid Network (FPN+PAN) are integrated into the Neck part and a Simplified Spatial Pyramid Pooling Fast (SimSPPF) module is introduced to realize the integration of local features and global features. Secondly, multiple Spatial and Channel Squeeze-and-Excitation (scSE) attention mechanisms are introduced in the Neck part to make the model pay more attention to the channels containing important feature information. The original six detection heads are reduced to four to improve the inference speed of the network. In order to improve the recognition performance of occluded and overlapping targets, DIoU-NMS was used to replace the original non-maximum suppression (NMS). Furthermore, the channel pruning strategy is used to reduce the unimportant weight matrix of the model, and the knowledge distillation strategy is used to fine-adjust the network model after pruning, so as to ensure the detection accuracy. The experimental results show that the parameter number of the proposed model is reduced from 26.15 M to 0.61 M, the computational load is reduced from 118.95 G to 1.49 G, and the mAP is increased from 96.8% to 98%. Compared with other models, the proposed model not only guarantees the detection accuracy of the algorithm, but also greatly reduces the model volume, which provides support for the realization of visible light insulator target detection based on edge intelligence.

## 1. Introduction

Insulators are essential components in power transmission lines, primarily functioning to provide electrical insulation and mechanical anchorage. However, due to their prolonged exposure to outdoor environments and the effects of overvoltage, they are susceptible to damage, which can impair the stable operation of the transmission lines and potentially lead to large-scale power outages, resulting in significant economic and social losses [1]. Therefore, to ensure the secure operation of power transmission lines, conducting the safety inspections of these lines is imperative. Conventional inspection methods entail visual examination by human operators or the use of specialized equipment to detect defects in insulators. Nonetheless, such approaches are prone to missing defects, making erroneous judgments, and are inefficient [2]. With the ongoing development of deep learning and drone technologies, they have come to play a crucial role in the safety inspections of power transmission lines [3]. Initially, drones equipped with high-definition cameras are utilized to conduct aerial photography of insulators on power transmission lines, capturing high-resolution images. Subsequently, deep learning techniques are employed to accurately identify and analyze defects within these images of the insulators.

In the realm of insulator object detection in power transmission lines, numerous scholars have conducted related research. The two-stage target detection algorithms include R-CNN [4], Fast R-CNN [5], and Faster R-CNN [6], which show an excellent performance and high detection accuracy in the field of target detection. Zhao [7] et al. present an improved Faster R-CNN-based method for insulator object detection, which achieves the precise detection of insulators under various aspect ratios, scales, and conditions of mutual occlusion by enhancing the anchor generation method within the Region Proposal Network (RPN) and refining the non-maximum suppression (NMS) strategy. Haijian [8] et al. propose a transmission line insulator detection method based on an improved Faster R-CNN, substituting the original VGG-16 network with a deeper Resnet-50 network and incorporating attention mechanism modules, and the target detection accuracy is improved by 1.63%, albeit at a cost of a slower detection speed. To address the issue of the slow recognition speed inherent in R-CNN series algorithms, Redmond et al. [9] introduced the YOLO (You Only Look Once) series of algorithms, which, as a type of single-stage object detection algorithm, offer a high detection speed but compromise on the detection precision to some extent. Juping [10] et al. proposed an overhead power transmission line object detection method based on an improved YOLOv5, which enhances the detection accuracy of small objects by incorporating larger-scale detection layers and skip connections into the algorithm. In addition, a small object-enhanced Complete Intersection over Union (CIoU) is put forward as the loss function of the bounding box regression. And pruning methods are adopted to lighten the model. The results indicate that this method achieves a 4% increase in the detection accuracy, a 58% reduction in the model size, and a 3.3% improvement in the detection speed. Wang [11] et al. introduced an insulator defect detection method based on ML-YOLOv5, in which the depthwise separable convolutions is employed as the backbone feature extraction network and the feature fusion module is improved by adopting an Enhanced Feature Pyramid Network (MFPN), and utilizing YOLOv5m as a teacher model for knowledge distillation. The experimental outcomes demonstrate that this algorithm boasts high detection accuracy and rapid detection speeds. In addition, Liu [12] et al. proposed a multi-scale detection algorithm, the SSD (Single-Shot MultiBox Detector) algorithm, which overcomes the shortcomings of the R-CNN series and YOLO series and has advantages in speed and accuracy. Xuan [13] et al. presented an improved SSD for the online detection of Insulators and Spacers Based on a UAV System. This approach utilizes the lightweight MnasNet network as the feature extraction network to generate feature maps and employs two multi-scale feature fusion strategies to integrate multiple feature maps. The outcomes illustrate that the algorithm excels in both a high detection accuracy and fast detection speed; however, there remains room for further reductions in the algorithm’s size.

Prior to inspecting defects on insulators, a preprocessing stage is indispensable: the identification and localization of insulators within images through object detection techniques. This preliminary step lays the groundwork for subsequent defect detection, enabling the system to focus its analysis on areas of potential defects in insulators, thereby enhancing the overall efficiency and accuracy of the inspection process. In pursuit of a balance between the detection accuracy, recognition speed, and a smaller algorithmic footprint, numerous scholars have adopted more lightweight convolutional neural network models; however, these still entail substantial computational loads and parameter counts [14]. Against this backdrop, this paper proposes a lightweight visible-light insulator object detection algorithm based on an improved SSD. First, improvements are made to the base model to enhance its detection accuracy, followed by pruning operations to achieve model lightweighting. To mitigate the decline in precision typically associated with pruning, knowledge distillation is employed to fine-tune the lightweight model. Ultimately, the performance of the proposed algorithm is validated using a visible-light insulator dataset, with a comparative analysis against classic object detection algorithms to confirm the efficacy of the improvement strategies outlined herein. The algorithm proposed in this paper can effectively solve the problem that the detection accuracy of the algorithm is not high in the insulator target detection task and the algorithm is too large to be deployed to a mobile terminal, such as the UAV.

## 2. SSD Network Model

The SSD object detection algorithm is characterized by multi-prediction layers and multi-scale features [15]. Its network architecture can be divided into three parts: First, the base network utilizes the VGG-16 structure to extract multi-scale feature information from the target. Second, auxiliary convolutional layers are connected to the final feature map of VGG-16, constructing deeper output layers for object detection. Third, the prediction convolutional layers obtain feature information from the feature maps and utilize NMS to derive the detection results [16]. The model architecture of the SSD algorithm is depicted in Figure 1. The input image size for SSD is 300 × 300 pixels, with VGG-16 serving as the feature extraction layer. Through six convolutional layers, it constructs multi-scale detection layers to capture feature information at various scales including 38 × 38, 19 × 19, 10 × 10, 5 × 5, 3 × 3, and 1 × 1, forming a multi-scale feature extraction network [17]. By setting prior boxes on feature maps of different depths and resolutions, and performing category prediction and location refinement for each prior box boundary, objects are precisely matched. The fact that different convolutional layers in a CNN have distinct receptive fields enables the network to effectively recognize targets of different sizes. Ultimately, the network computes the coordinates and class of candidate boxes through regression [18]. In the SSD algorithm, the formula for calculating the anchor box scales corresponding to each feature map is shown in Equation (1). Here, $S_{m}$ denotes the scale of candidate boxes for the *m*-th feature map; $S_{max}$ represents the maximum scale of candidate boxes, typically set at 0.9; $S_{min}$ signifies the minimum scale of candidate boxes, usually set to 0.2; and r denotes the total number of feature maps. The main symbols and their meanings are shown in Table 1.
(1)$$S_{m}=S_{min}+\frac{S_{max}-S_{min}}{r-1},\;m\in \left\{1,\;2,\;\ldots ,\;r\right\}$$

## 3. Improved SSD Network Model

This paper proposes the following improvements to the SSD network model based on the actual characteristics of power transmission line insulator images, with the enhanced SSD network structure illustrated in Figure 2.

(1)The original feature extraction network, VGG-16, is replaced with a lightweight Ghost Module network to initially achieve model lightweighting.(2)The Neck part of the SSD network adopts an FPN+PAN structure to enhance feature extraction capabilities. To facilitate the fusion of local and global features, a SimSPPF structure is introduced at each input end of the Neck.(3)Multiple Spatial and Channel Squeeze-and-Excitation (scSE) attention mechanism modules are incorporated into the Neck section, enabling the network to better focus on channels containing critical feature information while preserving positional information of feature layers.(4)The original six detection heads are reduced to four to accelerate the network’s inference speed. To improve the recognition of occluded and overlapping objects, DIoU-NMS replaces the conventional non-maximum suppression.(5)Channel pruning strategies are employed to eliminate unimportant weight matrices, further lightweighting the constructed network model and achieving model compression objectives.(6)To mitigate the impact of channel pruning on detection accuracy, knowledge distillation is applied to fine-tune the lightweight network model, ensuring detection precision is maintained.

### 3.1. Feature Extraction Network

The original SSD network model employs VGG-16 as its feature extraction backbone, which comprises a stack of numerous convolutional and pooling layers, leading to a deep network architecture. However, this model is characterized by a substantial number of parameters, necessitating longer training times and posing significant challenges in the tuning process, thereby hindering its deployment on mobile devices. Consequently, in pursuit of maintaining the detection performance while reducing the model size, this study eschews the VGG-16 network in favor of adopting a lightweight Ghost Module to construct the primary feature extraction backbone.

The Ghost Module functionally substitutes conventional convolution [19], capable of generating an equivalent number of feature maps to standard convolutional layers through a two-step process. Initially, a 1 × 1 convolution with fewer output channels is employed to perform dimensionality reduction, thereby creating a condensed feature map from the input feature layer. Subsequently, depthwise separable convolution is applied to this condensed map to yield similar feature maps. Finally, by concatenating the condensed feature map with its corresponding similar feature maps, an output feature map is attained, mirroring the structure of those produced by standard convolutions. The Ghost Module’s convolution operation encompasses two primary components. The first part involves obtaining intrinsic feature maps through conventional convolutional operations. If the input image size is *h* × *w* × *c*, the computational cost of this part is $h\times w\times c\times m\times w^{\prime }\times h^{\prime }$. The other part employs a simple linear operation to generate multiple feature maps, as illustrated by Equation (2), where depthwise separable convolution is applied to the original features. Each channel feature map, ${y^{\prime }}_{i}$, undergoes a linear operation, $\Phi_{i,\;j}$, to produce ghost feature maps.
(2)$$y_{ij}=\Phi_{i,\;j}({y^{\prime }}_{i}),\;\forall i=1,\;\ldots ,\;m,\;j=1,\;\ldots ,\;s$$

The theoretical acceleration ratio of replacing conventional convolutional modules with the Ghost Module is given by Equation (3), where d×d denotes the size of the linear operation kernel, which is comparable in magnitude to k×k and s<<c. Consequently, Equation (3) can be approximated by Equation (4), indicating that the Ghost Module entails significantly fewer parameters and computational costs compared to standard convolutions.
(3)$$r_{s}=\frac{n\cdot h^{\prime }\cdot w^{\prime }\cdot c\cdot k\cdot k}{\frac{n}{s}\cdot h^{\prime }\cdot w^{\prime }\cdot c\cdot k\cdot k+(s-1)\cdot \frac{n}{s}\cdot h^{\prime }\cdot w^{\prime }\cdot d\cdot d}\approx \frac{s\cdot c}{s+c-1}\approx s$$
(4)$$r_{c}=\frac{n\cdot c\cdot k\cdot k}{\frac{n}{s}\cdot c\cdot k\cdot k+(s-1)\cdot \frac{n}{s}\cdot d\cdot d}\approx \frac{s\cdot c}{s+c-1}\approx s$$

### 3.2. Feature Fusion Network

In order to solve the influence of different target sizes on the model detection accuracy caused by the change in shooting angle in the process of transmission line insulator image acquisition, the FPN+PAN structure is integrated into the Neck part of the SSD network to enhance feature extraction capabilities, particularly catering to objects of diverse scales [20]. Furthermore, to facilitate the fusion of local and global features, both the SimSPPF structure and scSE attention mechanism modules are introduced at the inputs of the Neck [21]. The FPN+PAN module is depicted in Figure 3, wherein the Feature Pyramid Network (FPN) structure performs upsampling from higher to lower dimensions of the backbone network’s outputs, thereby capturing strong semantic information. Conversely, the Path Aggregation Network (PAN) structure conducts downsampling from lower to higher dimensions, acquiring robust location information across various scales. Ultimately, these features are concatenated across dimensions, enabling the superior recognition of objects across different scales.

In the process of feature extraction from power transmission line insulator images, issues arise due to inconsistencies in image scales and distortions caused by operations such as resizing, cropping, and grayscale transformations. To circumvent these issues, this study integrates the SimSPPF module at the input end of the Neck section, facilitating the fusion of multi-scale insulator feature maps and global feature maps. The structure of the SimSPPF module is illustrated in Figure 4. This module processes the input data sequentially through several Maxpool layers with 5 × 5 kernel sizes. Notably, the combined outputs of two sequential 5 × 5 Maxpool layers equate to that of a single 9 × 9 Maxpool layer, and similarly, the combined output of three sequential 5 × 5 Maxpool layers matches that of a single 13 × 13 Maxpool layer. Consequently, the SimSPPF structure requires only three 5 × 5 convolution kernels to achieve the integration of local and global features, thereby enhancing computational efficiency and reducing computational overhead. Moreover, the SimSPPF module employs the ReLU activation function to expedite network inference, further boosting the detection efficiency.

To enhance the SSD network model’s capability in capturing and focusing on critical features in insulator images while preserving positional information in feature layers, this study incorporates multiple scSE attention mechanism modules into the Neck section [22]. As depicted in Figure 5, the scSE attention mechanism module is comprised of a parallel combination of a spatial squeeze–excitation (sSE) module and a channel squeeze–excitation (cSE) module. The sSE module reduces the channel information in feature maps to perform dimensionality reduction, and the compressed feature maps are then normalized by the sigmoid function σ(⋅) to obtain important spatial information, thereby invigorating key spatial features and increasing focus on crucial channel features. Meanwhile, the cSE module adjusts feature maps based on feature correlations across different channels. It compresses the feature map of size H × W × C through global average pooling, followed by activation through the ReLU function δ(⋅) and sigmoid normalization σ(⋅) to derive the importance of channel features, thereby enhancing attention to vital features [23].

When UAVs capture images of insulators, variations in the shooting angles often lead to the occlusion of the insulators in the photographs. To enhance the recognition of the occluded objects, this paper adopts DIoU-NMS in place of the conventional NMS technique [24]. The definition of DIoU-NMS is outlined by the following formula:(5)$$s_{i}=\left\{\begin{array}{ll} s_{i} & PIoU-RDIoU(M,B_{i})<\varepsilon \\ 0 & PIoU-RDIoU(M,B_{i})\geq \varepsilon \end{array}\right.$$
(6)$$R_{DIoU}(M,B_{i})=\frac{\rho^{2}(M,\;B_{i})}{c^{2}}$$

In the equation, *M* represents the prediction box with a higher prediction score, $B_{i}$ denotes the other prediction boxes, ρ is the Euclidean distance between *M* and $B_{i}$, and *c* is the diagonal distance of the smallest enclosing rectangle covering both *M* and $B_{i}$. DIoU-NMS effectively determines whether two overlapping boxes belong to the same object and efficiently suppresses bounding boxes. Compared to ground-level natural perspectives, the overlap rate of insulators is lower when viewed from a UAV perspective; hence, a small threshold ε is employed in this study to enhance the accuracy of the SSD algorithm in detecting insulator targets [25].

### 3.3. Model Compression and Fine-Tuning

To mitigate the reliance of the SSD network model on computational power, storage space, and other resources of edge intelligent terminals, this study employs channel pruning strategies to compress the enhanced SSD network model [26]. In order to accelerate the model convergence, *BN* layers are introduced after convolutional layers. The *BN* layers process the input data through shift and scaling parameters, normalizing the outputs of each convolutional layer within a reasonable range, as depicted in Equations (7) and (8).
(7)$$Y=BN(X)=\gamma \frac{X-\mu }{\sqrt{\sigma^{2}+\varepsilon }}+\beta$$
(8)$$Y=\underset{\gamma \to 0}{\lim}\gamma \frac{X-\mu }{\sqrt{\sigma^{2}+\varepsilon }}+\beta =\beta$$

In this context, X represents the input to the *BN* layer and *Y* denotes the output from the BN layer. σ and μ are, respectively, the variance and mean computed over a mini-batch for the *BN* layer. β serves as a bias compensation in the normalization process, while γ acts as a scaling factor post-normalization, signifying the importance of channels. ε is a small non-zero constant to prevent division by zero. When γ approaches zero, the activation function following the BN layer maps the channel inputs to smaller output values [27], suggesting that the corresponding channel contributes minimally to the *BN* layer’s output. Consequently, this redundant channel can be pruned, leading to a lightweight network architecture.

During conventional training, the model’s loss function does not incorporate γ, resulting in a post-training distribution of γ that tends towards a normal distribution with most values close to 1, making pruning of the model challenging. To identify redundant channels, γ needs to be incorporated into the loss function for sparsification training, with an L1 regularization imposed on γ to drive the model parameters towards structured sparsity, thereby facilitating the identification of crucial channels [28]. The modified loss function is expressed as Equation (9).
(9)$$L=\sum_{\left(x,\;y\right)}l(f(x,W),y)+\lambda \sum_{\gamma \in \Gamma }g(\gamma )$$

In the equation, the first summation represents the loss function of the conventionally trained model, while the second summation denotes the L1 regularization penalty term. *L* signifies the loss function for sparse training, with Γ encompassing all prunable channels. The function g(γ) embodies the L1 regularization, here defined as $g(\gamma )=\left|\gamma \right|$. Initially, the model undergoes standard training. Following this, the well-trained model is subjected to sparse training via the loss function *L*, promoting sparsity. Upon the completion of the sparse training, the relevance γ of redundant channels diminishes towards zero, thus accomplishing model lightweighting.

To address the degradation in performance resulting from model pruning, this study employs knowledge distillation to fine-tune the model post-channel pruning [29]. Leveraging transfer learning, complex teacher networks guide simpler student networks, migrating knowledge to the student model. In this work, YOLOv5 is selected as the teacher network, as depicted in Figure 6, with the student network adopting a hint-based learning strategy to imbibe pertinent features from the teacher network [30]. To counteract the imbalance between insulator targets and the background during object recognition, a weighted cross-entropy loss is employed in the knowledge distillation network. In pursuit of further enhancing the network performance, the regression outputs from the teacher network are used as upper bounds, ensuring that the student network is not penalized even if it outperforms the teacher, thus fostering an environment conducive to learning without constraints [31]. This methodology promotes the preservation and enhancement of critical detection capabilities in the distilled, lightweight model [32,33].

## 4. Experimental Results and Analysis

### 4.1. Experimental Environment

The experiments reported herein were conducted on a 64-bit Windows 11 operating system, utilizing the deep learning framework PyTorch. The detailed configuration of the experimental environment is presented in Table 2.

### 4.2. Datasets and Training

To validate the effectiveness of the algorithm presented in this paper, an open-source visible light insulator dataset was selected. In order to enhance the generalization capability of the model, data augmentation techniques were applied to a portion of the sample images, thereby increasing the diversity of the dataset. These techniques included image cropping, stitching, color space transformation, resizing, among others. The augmented dataset was then annotated using the LabelImg (version 1.8.6) tool, with insulators in the dataset labeled consistently as ‘Insulator’. The annotation format adhered to PascalVOC standards, ensuring uniformity across similar objects. Upon the completion of the labeling process, XML files were generated and stored within a label directory, each corresponding to an annotated image. The annotated files and their respective dataset images were meticulously paired and subsequently split into training and testing sets at an 8:2 ratio. The contents of the segmented annotation files were further converted into a training text and testing text, formatted according to predefined specifications, to facilitate model training.

The training set was fed into an improved SSD network model, with the maximum learning rate initialized at 0.01 and decreased to a minimum of 0.0001 throughout training. A batch size of 16 was employed to balance the computational efficiency and memory utilization. The model underwent 300 iterations of training, during which the optimal weights were saved for future deployment. The input image resolution was standardized to 416 × 416 pixels to accommodate the architecture’s requirements and enhance feature extraction.

The convergence curve of the training loss for the enhanced SSD model is illustrated in Figure 7, demonstrating the model’s learning progression and stability over the course of the training epochs. The pseudo-code of the proposed model (Algorithm 1) is presented in the following form:
**Algorithm 1**: pseudo-code of the proposed modelInput: An image to be detectedOutput: An image with detection results1:     Resize the input image to 416 × 416 and normalize it.2:     Pass the processed image through the backbone network to extract features.3:     Feed the extracted features into the network model (backbone, neck, and head) to obtain candidate bounding boxes.4:     For each candidate bounding box:               Perform classification and bounding box regression;               Decode the regression results to determine the final position of the bounding box;               Apply DIoU-NMS to filter out overlapping detections;               Map the detection result onto the original image.5:     Return the image with the overlaid detection results.

### 4.3. Evaluating Indicator

This paper evaluates the enhanced SSD algorithm as using metrics including mean Average Precision (*mAP*), Precision (*P*), Recall (*R*), Frames Per Second (FPS), the number of parameters, and Floating-point Operations Per Second (FLOPs). A higher *mAP* indicates greater detection accuracy, while larger numbers of parameters and higher computational loads signify a bulkier algorithm. Particularly, smart edge devices impose stringent constraints on the model size in terms of both the parameter count and computational demands. The term mAP@0.5 signifies the average precision across all classes when the Intersection over Union (IoU) threshold is set to 0.5, reflecting the trend of precision as recall varies. *R* measures the proportion of true positive samples correctly identified, thereby gauging the extent of missed detections. *P*, on the other hand, assesses the fraction of predicted positive samples that are indeed true positives, indicating the rate of false alarms [34]. FPS quantifies the speed of detection, with a higher FPS translating to faster detection. FLOPs is used to evaluate the computational complexity of the model. The relevant formulas are as follows, where *Tp* denotes true-positive predictions, *Fp* represents false-positive predictions (negative samples incorrectly labeled as positive), and *Fn* indicates false-negative predictions (positive samples mislabeled as negative).
(10)P=Tp/(Tp+Fp)
(11)R=Tp/(Tp+Fn)
(12)$$AP=\int_{0}^{1}P(R)dR$$
(13)$$mAP=\frac{\sum_{i=1}^{N}APi}{N}$$

### 4.4. Ablation Experiment

To verify the performance of the improved SSD model in detecting insulator targets, ablation experiments were conducted to compare the original SSD network with the model proposed in this paper. The setup for these ablation experiments is summarized in Table 3, where “√” denotes the inclusion of a module, and “×” indicates its exclusion. The outcomes of these ablation experiments are presented in Table 4.

According to Table 3, comparing Model A with the original SSD algorithm reveals that after replacing the SSD’s backbone feature extraction network, VGG-16, with the Ghost Module, the model size is reduced by 80.6%, albeit at the expense of a decrease in the detection accuracy. This confirms that while the Ghost Module employs fewer parameters, leading to a smaller model size, it also has an adverse effect on the detection precision. Comparing Models A, B, and C illustrates that the introduction of the SimSPPF structure and scSE attention mechanism leads to negligible changes in the model size, but improves the accuracy, validating that these enhancements strengthen the model’s comprehension and processing of input data, thereby enhancing the detection accuracy. The comparison between Models C and D shows that Model D, which incorporates the FPN+PAN structure and undergoes channel pruning and knowledge distillation, achieves a 91.3% reduction in model size while improving the detection accuracy by 2.4%. This evidence supports the notion that the FPN+PAN structure, along with knowledge distillation and channel pruning, significantly reduces the model size while effectively boosting the detection accuracy.

### 4.5. Comparison of Different Algorithm Effects

To validate the efficacy of the proposed algorithm, comparative experiments were conducted against classical object detection algorithms using the same dataset. The experimental environment was consistent with that of the ablation experiments, with all models trained for 300 iterations. The results of these comparative experiments are summarized in Table 5.

According to the results in Table 5, it can be seen that the algorithm in this paper has the smallest computational and parameter requirements. Compared with algorithms such as YOLOv3, YOLOv5, and Faster RCNN, the model has significantly reduced computational and parameter requirements, with an average accuracy of 98%, slightly lower than YOLOv5 and Faster RCNN. Compared with lightweight algorithms such as Ghost-YOLOv3 and YOLOv3-Tiny, our algorithm has a lower parameter and computational complexity, and higher detection accuracy. In summary, while maintaining high accuracy, the algorithm proposed in this article has the lowest number of model parameters and computational complexity.

Figure 8 illustrates the detection results of various algorithms. The figure indicates that the proposed algorithm and Faster-RCNN achieve the highest accuracy in detecting insulators in Figure 8a. In the insulator images depicted in Figure 8b,c, the original SSD, YOLOv3-tiny, and YOLOv3 all exhibit varying degrees of missed detections. Faster-RCNN achieves the highest detection accuracy; however, Table 4 reveals that it is larger in size and slower in detection speed. Upon a comprehensive comparison, the proposed algorithm shows a superior overall performance, confirming the effectiveness of the improvement strategies outlined herein.

## 5. Conclusions

Addressing the challenge of balancing the detection accuracy with the model size in power transmission line insulator inspection algorithms, which hampers their deployment on embedded devices, this paper presents a lightweight insulator target detection model based on improved SSD. The main contents of this paper are as follows:

(1) Through the introduction of the lightweight Ghost Module network, the initial lightweight of the model is realized. (2) By introducing a SimSPPF structure and FPN+PAN structure, the fusion of the local and global features of the model is promoted, and the feature extraction capability of the model is enhanced. (3) By introducing multiple spatial and channel squeeze incentive (scSE) attention mechanism modules, the model’s ability to focus on key features is improved. (4) By reducing the number of detection headers and introducing the DIoU-NMS mechanism, the detection speed is improved, and the model’s recognition ability of occluded and overlapping targets is improved. At the same time, through channel pruning and knowledge distillation, the number of model parameters is further reduced and the accuracy of model detection is improved.

The major conclusions in this paper are listed as follows.

By using the lightweight Ghost Module network to replace the original feature extraction network, VGG-16, the model size is reduced by 80.6%, indicating that the lightweight module can effectively reduce the model size. However, the model detection accuracy is also reduced.

With the introduction of the SimSPPF structure and scSE attention mechanism, although the model size is not significantly reduced, the detection accuracy is effectively improved, which proved that the SimSPPF structure and scSE attention mechanism can effectively improve the model’s ability to understand and process input data, thus effectively improving the detection accuracy.

Combined with the FPN+PAN structure, channel pruning and knowledge distillation were performed on the model. The number of parameters decreased by 91.3%, from 7.04 M to 0.61 M, and the detection accuracy increased by 2.4%, from 95.6% to 98.0%, which verified the effectiveness of the improvement measures taken in this paper.

Compared with other models, the proposed model has the lowest number of parameters while maintaining high detection precision, and other parameters have been effectively improved, which shows that the improvement strategy in this paper can significantly improve the model performance. The model in this paper not only ensures the detection accuracy, but also minimizes the consumption of hardware resources, and can meet the requirements of the deployment and application on edge intelligent terminals. However, the resource utilization and energy consumption of the algorithm have not been deeply studied in this paper. In the future, how to reduce the resource utilization rate and energy consumption of the algorithm at the edge intelligent terminal will be deeply studied, and the endurance time of the UAV can be further improved while ensuring the accuracy and efficiency of target detection and defect identification.

## Figures and Tables

**Figure 1 sensors-24-05910-f001:**
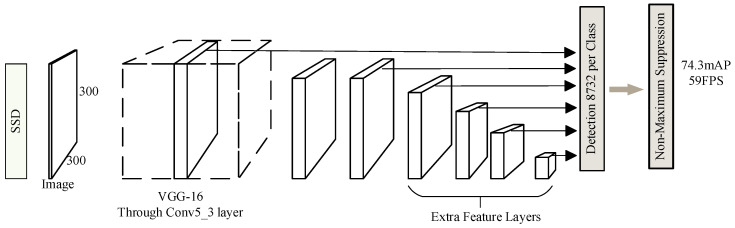
SSD algorithm model.

**Figure 2 sensors-24-05910-f002:**
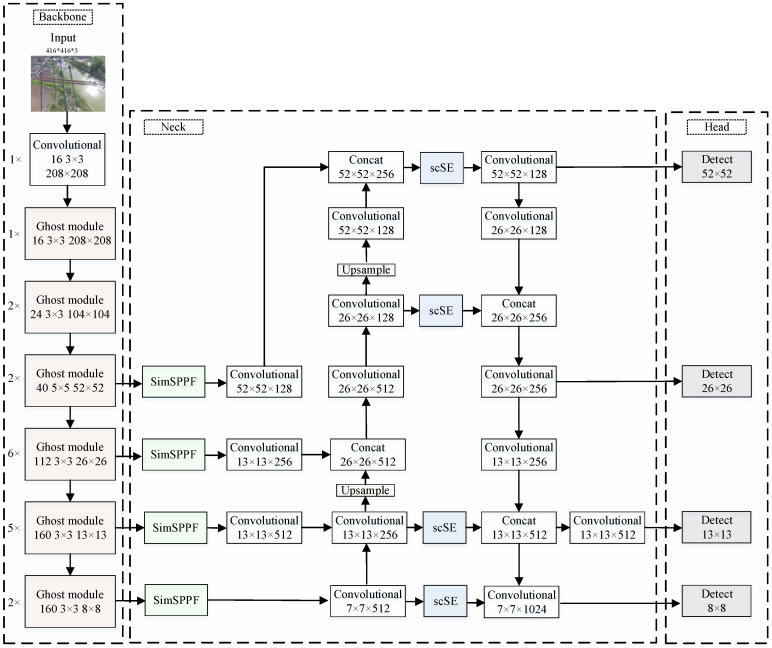
Improved SSD network model.

**Figure 3 sensors-24-05910-f003:**
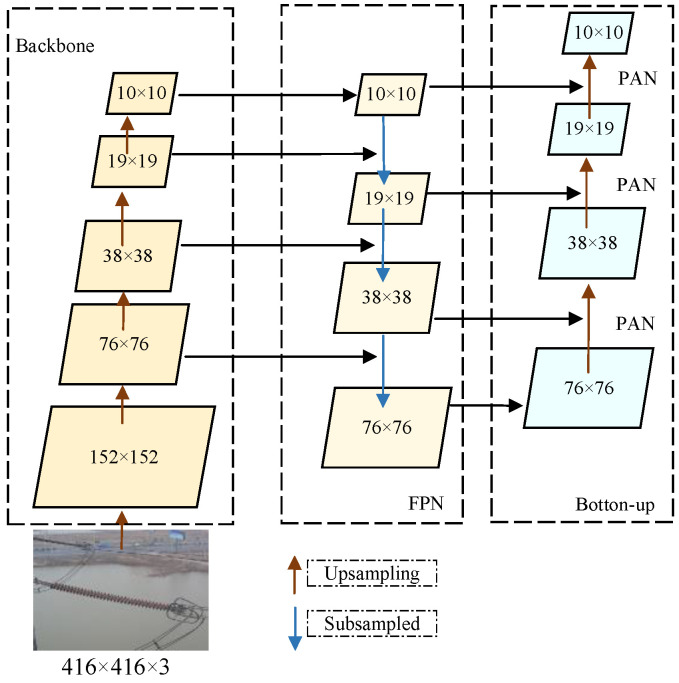
FPN+PAN Module.

**Figure 4 sensors-24-05910-f004:**
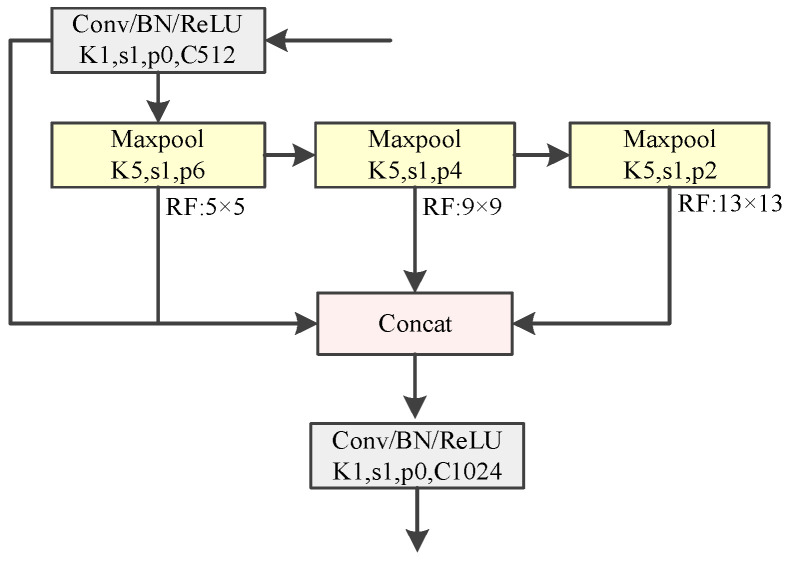
SimSPPF network structure.

**Figure 5 sensors-24-05910-f005:**
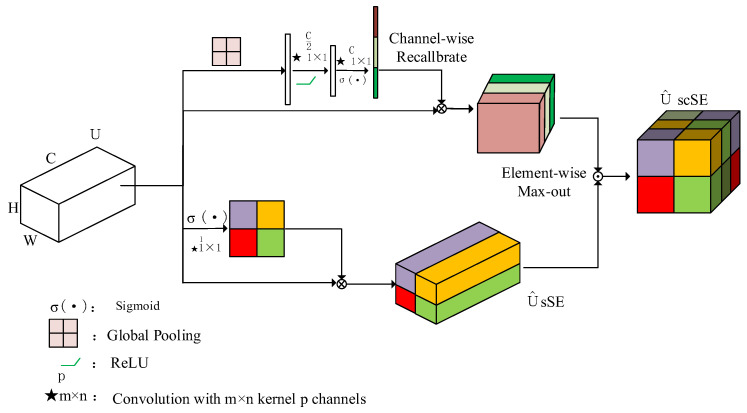
Attention mechanism model.

**Figure 6 sensors-24-05910-f006:**
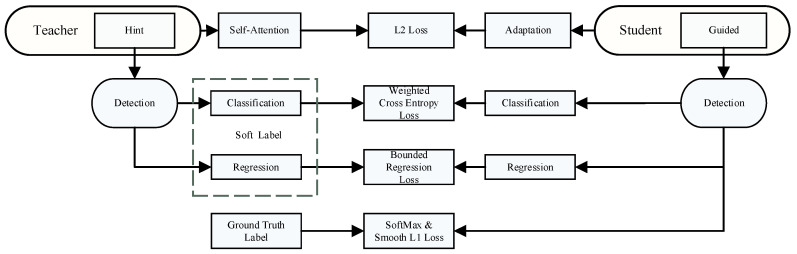
Knowledge distillation diagram.

**Figure 7 sensors-24-05910-f007:**
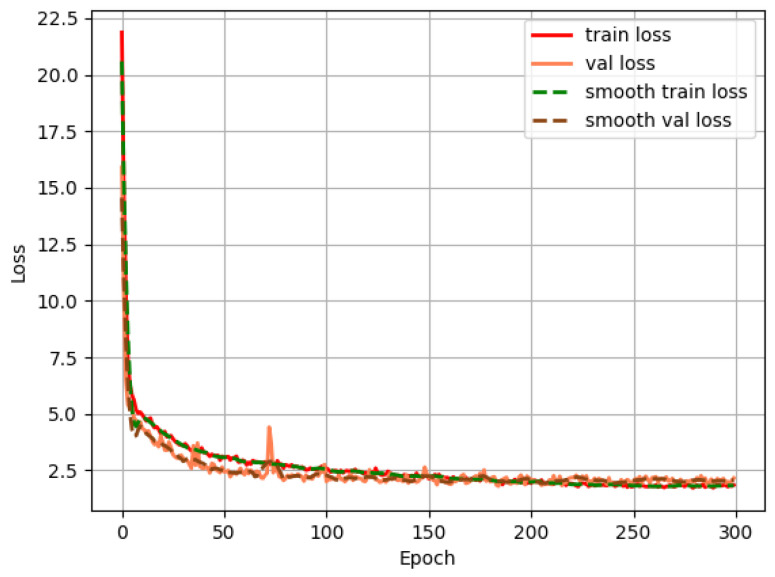
Training loss curve of improved SSD model.

**Figure 8 sensors-24-05910-f008:**
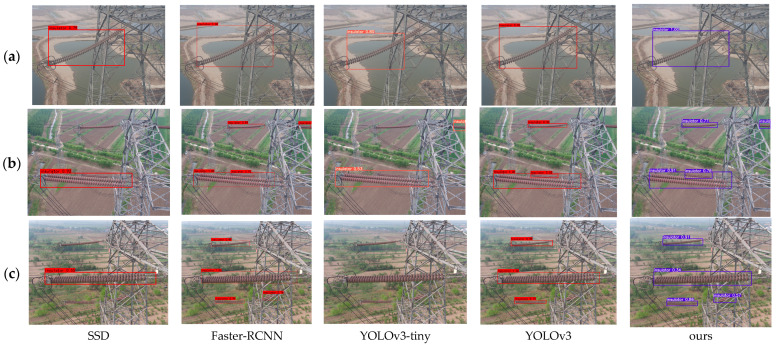
Detection results of insulator detection models on different data sets. (**a**) shows the results of different algorithms on a single insulator. (**b**) shows the detection results of different models when the insulators in the photos are incomplete and the scales are different. (**c**) shows the detection results of different models with different insulator scales.

**Table 1 sensors-24-05910-t001:** Main symbols and their meanings.

Symbol	Description
$S_{m}$	the scale of candidate boxes for the *m*-th feature map
$S_{max}$	the maximum scale of candidate boxes
$S_{min}$	the minimum scale of candidate boxes
r	the total number of feature maps
*h*	the picture height
*w*	the picture width
*c*	the picture length
${y\prime }_{i}$	channel feature map
$\Phi_{i,\;j}$	undergoes a linear operation
d×d, k×k	the size of the linear operation kernel
σ(⋅)	the compressed feature maps are then normalized by the sigmoid function
δ(⋅)	the ReLU function
*M*	the prediction box with a higher prediction score
$B_{i}$	the other prediction boxes
ρ	the Euclidean distance between *M* and $B_{i}$
ε	non-zero constant
X	the input to the *BN* layer
*Y*	the output from the *BN* layer
γ	represents the normalized scale factor
σ	the variance computed over a mini-batch for the *BN* layer
μ	the mean computed over a mini-batch for the *BN* layer
β	a bias compensation in the normalization process
Γ	encompassing all prunable channels
FPS	Frame Per Second
*Tp*	true positive predictions
*Fp*	false positive predictions
*Fn*	indicates false negative predictions

**Table 2 sensors-24-05910-t002:** Experimental environment configuration.

Category	Parameter
CPU	12th Gen Intel(R) Core(TM) i7-12700KF 3.6 GHz
Memory	32 G
GPU	NVIDIA GeForce RTX 3090Ti
GPU memory	24 G
OS	Windows 11
CUDA version	CUDA 11.0
cuDNN	cuDNN 7.6.5
Language	Python 3.6

**Table 3 sensors-24-05910-t003:** Ablation experimental design.

Models	VGG-16	Ghost Module	SimSPPF	scSE	FPN+PAN	Lightweight
SSD	√	×	×	×	×	×
A	×	√	×	×	×	×
B	×	√	√	×	×	×
C	×	√	√	√	×	×
D	×	√	√	√	√	√

**Table 4 sensors-24-05910-t004:** Ablation experimental results.

Models	Parameters	FLOPs	*P*	*R*	FPS f/s	mAP@0.5/%
SSD	26.15 M	118.95 G	0.95	0.81	132	96.8%
A	5.07 M	3.21 G	0.96	0.79	111	93.2%
B	5.20 M	3.38 G	0.97	0.77	101	94.8%
C	7.04 M	3.39 G	0.94	0.81	101	95.6%
D	0.61 M	1.49 G	0.78	1	67	98.0%

**Table 5 sensors-24-05910-t005:** Improved SSD algorithm compared with mainstream target detection algorithm.

Models	Lightweight	FPS/f/s	*P*	*R*	Parameters	FLOPs	mAP@0.5/%
SSD		132	0.95	0.81	26.15 M	118.95 G	96.8%
YOLOv3		71	0.80	0.98	61.52 M	65.60 G	96.9%
YOLOv5		94	0.94	0.99	47.06 M	115.92 G	98.2%
Faster-RCNN		22	0.70	1	137.10 M	370.21 G	98.6%
Ghost-YOLOv3	√	55	0.77	0.98	46.45 M	25.32 G	95.5%
YOLOv3-Tiny	√	142	0.62	0.81	8.67 M	5.49 G	73.8%
Ours	√	67	0.78	1	0.61 M	1.49 G	98.0%

## Data Availability

The raw data supporting the conclusions of this article will be made available by the authors on request.

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
