# Peer review of "Research on Lightweight Method of Insulator Target Detection Based on Improved SSD"

_sensors, 2024, doi:10.3390/s24185910_

Round 1

Reviewer 1 Report

Comments and Suggestions for Authors

To handle the challenge of balancing detection accuracy with model size in power transmission line insulator inspection algorithms, which hampers their deployment on embedded devices, this paper presents improvements to the SSD object detection algorithm.  It mainly involves the following aspects. (1) The lightweight Ghost Module network is used to replace the original feature extraction network VGG-16; (2) With the introduction of SimSPPF structure and scSE attention mechanism, although the model size is not significantly reduced, the detection accuracy is effectively improved, which proved that the SimSPPF structure and scSE attention mechanism can effectively improve the model's ability to understand and process input data, thus effectively improving the detection accuracy; (3)Combined with FPN+PAN structure, channel pruning and knowledge rectification were performed on the model. However, the following question should be solved.

(1) The article uses many parameters, making it challenging to remember their specific meanings. Organizing them in a table for unified representation would be a more effective approach.

 (2)Some Equations in the paper are shown incompletely. To improve the readability of the paper, please explain the meaning of each variable.

(3) In the paper, what is the time complexity for the proposed algorithm? It is better for the authors to give pseudocode and some explanations to the proposed algorithm such as the function of each step, the overall basic idea.

(4) The proposed algorithm in the paper considers some factors, such as energy consumption and response time.   Does the author consider more factors, such as QoS and resource utilization? Some published papers have addressed these questions. For example, An Automatically Layer-Wise Searching Strategy for Channel Pruning Based on Task-Driven Sparsity Optimization, Doi: 10.1109/TCSVT.2022.3156588; Doi: 10.1109/TGCN.2021.3067309.  They have also solved these problems.  It is better that the authors list a Table to make a comparison with these latest high quality paper in the “Related work” section. The authors should focus on the nolvety of the paper,  advantage,  disadvantage,  and evaluation factors.

(5) In the paper,  the authors consider the energy consumption factor. However, any power model is essential for energy-aware algorithms.   Do the authors consider the energy consumption models?  

(6) The quality of the Figures in the paper should be further improved in the revision.

(7)There are many spelling errors in the manuscript. A thoroughly spelling check is required in the revision. 

Author Response

Comment 1:(1) The article uses many parameters, making it challenging to remember their specific meanings. Organizing them in a table for unified representation would be a more effective approach.

Response 1:Agree. According to your suggestion, the authors have added a table in the body, which lists the relevant parameters and the specific meaning of the parameters, as shown in Table 1.

Comment 2:(2) Some Equations in the paper are shown incompletely. To improve the readability of the paper, please explain the meaning of each variable.

Response 2:Agree. According to your suggestions, the author has perfected the meaning of the variables involved in the formula in the paper to ensure the readability of the paper and the integrity of the formula. Meanwhile, the meaning of the variables is listed in the body of the explanation, and the explanation of the meaning of the variables is supplemented in Table 1.

Comment 3:(3) In the paper, what is the time complexity for the proposed algorithm? It is better for the authors to give pseudocode and some explanations to the proposed algorithm such as the function of each step, the overall basic idea.

Response 3:Agree. Time complexity refers to the computational amount of forward reasoning, measured by Floating point Operations Per Second (FLOPs). In the object detection algorithm model, there are special functions to calculate the time complexity. At the same time, according to your suggestion, the authors add the pseudo-code of the proposed algorithm in the paper and introduces the process of the algorithm, as shown in Algorithm1.

Comment 4:(4) The proposed algorithm in the paper considers some factors, such as energy consumption and response time.   Does the author consider more factors, such as QoS and resource utilization? Some published papers have addressed these questions. For example, An Automatically Layer-Wise Searching Strategy for Channel Pruning Based on Task-Driven Sparsity Optimization, Doi: 10.1109/TCSVT.2022.3156588; Doi: 10.1109/TGCN.2021.3067309.  They have also solved these problems. It is better that the authors list a Table to make a comparison with these latest high-quality paper in the “Related work” section. The authors should focus on the nolvety of the paper, advantage, disadvantage, and evaluation factors.

Response 4: At present, the performance parameters of the algorithm, including mPA, are compared in this paper. QoS and resource utilization are not evaluated. On the one hand, the algorithm in this paper runs on the edge intelligent terminal and does not involve the image transmission at the network level, so QoS is not measured. At the same time, in the testing process of the algorithm, the comparison of the performance indicators of the algorithm on the fixed configuration hardware equipment is considered, and no long-term resource utilization evaluation is carried out. In addition, the resource utilization evaluation process of the hardware equipment involves not only image processing, but also many other aspects, including the interaction with the image acquisition device, such as the control of the device. This part also creates resource consumption. Some of the references you provide, the author has carried out a detailed reading, has a strong reference significance. In the future, the author plans to conduct a comprehensive evaluation of the long-term resource utilization rate of the algorithm in the actual application process, and continuously reduce the resource utilization rate of the algorithm on the premise of ensuring that the hardware configuration of the edge intelligent terminal meets the application requirements, so as to achieve the reduction of comprehensive energy consumption.

Comment 5:(5) In the paper, the authors consider the energy consumption factor. However, any power model is essential for energy-aware algorithms.   Do the authors consider the energy consumption models?  

Response 5:In the process of studying the performance of the algorithm, the detection speed and memory consumption of the algorithm have been evaluated, but the energy consumption has not been evaluated in depth. Considering the need to deploy applications in the on-board edge intelligent terminal of the UAV, lower energy consumption can make the UAV have a longer endurance, but the energy consumption itself may be affected by many factors. Considering that the paper focuses on the performance of the algorithm itself, the author will carry out further research on the energy consumption of the algorithm in the future combined with the comments of the reviewer.

Comment 6:(6) The quality of the Figures in the paper should be further improved in the revision.

Response 6:Agree. According to your suggestions, the authors have improved the quality of the Figures in the paper.

Comment 7:(7) There are many spelling errors in the manuscript. A thoroughly spelling check is required in the revision.  

Response 7:Agree. According to your suggestions, the authors have checked and improved the spelling in the article as a whole.

Thank you very much for these good suggestions. The authors have made a lot of modifications on this paper according to your suggestions, so that the paper can be improved.

Reviewer 2 Report

Comments and Suggestions for Authors

The paper presents a novel approach to insulator target detection using an improved SSD algorithm, which is a significant contribution to the field of power transmission line maintenance. I do, however, have some comments about the paper, which I believe would strongly improve the paper.

1. The authors should clearly describe their motivation to design this model. As shown in this manuscript, the proposed method works like a stack of existing modules.

2. Please summarize the contributions of this work in the revised manuscript.

3. There are a few spelling and grammar issues that need to be addressed, such as "Pytorch" which should be "PyTorch".

4. I strongly recommend the authors add more references about object detection and knowledge distillation, which would make the paper more readable, such as: Surface defect detection competition with a bio-inspired vision sensor.

Author Response

Comment 1: 1. The authors should clearly describe their motivation to design this model. As shown in this manuscript, the proposed method works like a stack of existing modules.

Response 1: At present, UAV has been widely used in transmission line inspection work. Through the UAV equipped with visible light, infrared and ultraviolet equipment, it can achieve efficient detection of transmission line defects or faults. However, the existing defect or fault detection, on the one hand, need to manually consult the image for identification, there is a lack of low efficiency, and may lead to misjudgment or missing judgment. On the other hand, the analysis of the acquired images by high-performance workstations leads to the lag in the identification of defects or faults. In order to realize the real-time detection of defects or hidden dangers by UAV equipped with edge intelligent terminals, a lighter algorithm model needs to be developed before it can be deployed and applied in edge intelligent terminals. To meet this demand, a lightweight target detection model for transmission line insulators is proposed in this paper.

In the process of building the model, the author learned about some existing modules by reading a lot of literature. However, in the actual application process, not every module can meet the application requirements. In the fusion test of some modules, it is found that the performance improvement of the module is not obvious. At the same time, in the process of knowledge distillation and module pruning, some of the existing modules cannot be used directly. In this respect, we need to do some adaptive processing. Therefore, in the actual research process, some of the existing modules were partially adjusted, and a lot of tests were carried out to obtain a better model.

Comment 2: 2. Please summarize the contributions of this work in the revised manuscript.

Response 2: Agree. According to your suggestion, the contributions of this work have been summarized in the revised manuscript.

Comment 3: 3. There are a few spelling and grammar issues that need to be addressed, such as "Pytorch" which should be "PyTorch".

Response 3: Agree.  According to your suggestion, “Pytorch” has been changed into “PyTorch”.

Comment 4: 4. I strongly recommend the authors add more references about object detection and knowledge distillation, which would make the paper more readable, such as: Surface defect detection competition with a bio-inspired vision sensor.

Response 4: Agree. According to your suggestion, the authors add more references about object detection and knowledge distillation, such as reference[30-33].

Thank you very much for these good suggestions. The authors have made a lot of modifications on this paper according to your suggestions, so that the paper can be improved.

Round 2

Reviewer 1 Report

Comments and Suggestions for Authors

All of my concerns have been addressed.